# A CBCT Investigation of the Sella Turcica Dimension and Sella Turcica Bridging in Different Vertical Growth Patterns

**DOI:** 10.3390/jcm12051890

**Published:** 2023-02-27

**Authors:** Shiyi Yan, Sheng Huang, Zuping Wu, Ying Liu, Yanling Men, Xiuping Nie, Jie Guo

**Affiliations:** 1Department of Orthodontics, School and Hospital of Stomatology, Cheeloo College of Medicine, Shandong University & Shandong Key Laboratory of Oral Tissue Regeneration & Shandong Engineering Laboratory for Dental Materials and Oral Tissue Regeneration & Shandong Provincial Clinical Research Center for Oral Diseases, Jinan 250012, China; 2Stomatology Hospital, School of Stomatology, Zhejiang University School of Medicine, Clinical Research Center for Oral Diseases of Zhejiang Province, Key Laboratory of Oral Biomedical Research of Zhejiang Province, Cancer Center of Zhejiang University, Hangzhou 310006, China

**Keywords:** sella turcica, craniofacial growth, CBCT, malocclusion

## Abstract

This study aimed to compare the sella turcica dimensions and sella turcica bridging (STB) via cone-beam computed tomography in different vertical patterns and then analyze the link between the sella turcica and vertical growth patterns. The CBCT images of 120 skeletal Class I subjects (an equal proportion of females and males; mean age of 21.46 years) were divided into three vertical growth skeletal groups. Student’s *t* tests and Mann–Whitney U tests were used to assess the possible diversity in genders. The link between sella turcica dimensions and different vertical patterns was explored by one-way analysis of variance, as well as Pearson and Spearman correlation tests. The prevalence of STB was compared using the chi-square test. Sella turcica shapes were not linked to gender, but statistical differences were observed among different vertical patterns. In the low-angle group, a larger posterior clinoid distance and smaller posterior clinoid height, tuberculum sellae height, and dorsum sellae height were determined, and the incidence of STB was higher (*p* < 0.01). Sella turcica shapes were linked to vertical growth patterns, mainly involving the posterior clinoid process and STB, which could be used as an index to assess vertical growth trends.

## 1. Introduction

The saddle-shaped structure situated on the intracranial surface of the sphenoid bone is called the sella turcica, and it is three-dimensional. Its anterior margin is known as the tuberculum sellae, and the posterior margin, which surrounds the pituitary gland, is known as the dorsum sellae [1]. Its center point, the sella point, is of vital importance in orthodontic cephalometry landmarks and plays a crucial part in image analysis [2]. The anterior and posterior sections of the sella turcica are separately developed from neural crest cells and the para-axial mesoderm. Neural crest cells were found to have nothing to do with the notochord, while the para-axial mesoderm relied heavily notochordal induction, which has been confirmed previously [3,4]. Studies have shown that sella turcica shapes were relevant to the development of the pituitary gland and neural crest cells [5]. An abnormal pituitary gland affects the secretion of a growth hormone, which is not conducive to the development of bone and body. Moreover, the mutations of the homeobox genes in cranial neural crest cells may have an impact on the development of the dentition and midface through signaling conduction [6]. This suggests that the sella turcica is closely related to the growth and development of the craniofacial region.

Two factors showed the importance of evaluating the dimensions and abnormal shapes of the sella turcica: its important anatomical position and its common embryologic origin with a cranial base [2,7,8]. Studies found that the ligament between the anterior and posterior clinoid processes may initiate ossification, also known as sella turcica bridging (STB), which was regarded as a developmental abnormality [9]. Previous studies have proven the possible association between STB and multiple diseases such as intrasellar adenoma [10], Down syndrome [11], and so on. Some scholars have proposed that malocclusion may be related to changes in the sella turcica. Tepedino et al. [7,12] concluded that there were various sella turcica dimensions in different sagittal growth patterns. Studies have shown that the sella turcica in Class II subjects was smaller than in Class III subjects [13]. In addition, patients with STB were more likely to have a greater distal position in the mandible [14]. Malocclusion can be explained by the developmental changes in the maxilla, mandible, or both. According to embryological studies, they had significant similarities with the sella turcica [12]. The growth and location of the maxilla and mandible affect skeletal patterns, including sagittal and vertical skeletal patterns. Moreover, different vertical growth patterns affect orthodontic treatment decisions. Therefore, the correlation between the sella turcica and the vertical growth patterns needs further research and analysis. Currently, there are a few studies on the different vertical growth patterns. According to the study by Perović, differences were found only for sella depth [15], which provides limited information.

Recently, some scholars have advanced the idea that different imaging methods and judgment methods of STB may affect the research results [16]. There were statistical differences in the measurements of the sella turcica between lateral cephalograms (LCRs) and cone-beam computed tomography (CBCT) [17,18], and the superposition of the overlapping structure of the sella turcica in two-dimensional imaging increased the false-positive rate of STB [19,20]. Tassoker et al. argued that only three-dimensional imaging such as CBCT could produce a more exact characterization of the sella area [21]. Previous studies, based on two-dimensional imaging, were filled with uncertainty. Thus, different skeletal growth patterns of the sella turcica are yet to be evaluated. We need more quantitative and objective research, using standard and highly sensitive methods.

Above all, this study aimed to (1) measure sella turcica shapes in subjects after the pubertal period (including different genders and different vertical patterns) and (2) to analyze the relationship between the sella turcica and vertical growth patterns by comparing differences within examined groups. The null hypothesis was that no relation could be found between sella turcica shapes and different vertical growth patterns.

## 2. Materials and Methods

### 2.1. Sample Selection

Our study was conducted in the Department of Orthodontics, School and Hospital of Stomatology, Cheeloo College of Medicine, Shandong University. The subjects included 120 individuals (60 women and 60 men) who underwent CBCT for joint discomfort, as well as the extraction of complexes impacted by teeth and other pathologies. This study was ratified by Shandong University Medical Research Ethics Committee (No: 20220910). Consent forms were signed by all subjects and the appropriate person with parental authority.

According to the following criteria, subjects were selected randomly if they: (a) were older than 16 years and if their sella turcica was basically developed, (b) were systemically healthy and had no craniofacial syndrome, (c) had no history of orthodontic or orthognathic surgical treatment, and (d) had Class I sagittal growth patterns evaluated by an ANB angle between 0.7 and 4.7°. They were excluded if they had (a) a history of trauma and treatment in the craniofacial region or (b) a history of long-term drug use affecting bone development.

The CBCT of the selected subjects were collected, and then LCRs were obtained from CBCT as reconstructed LCRs by dolphin software. Depending on the SN-MP, FH-MP, Y axis angle, and FHI [22,23], subjects were allocated to low, normal, or high angles. Thus, three clear-cut groups were generated, and each group had 40 subjects and an equal proportion of genders.

Low: SN-MP < 27.3°, FH-MP < 25.5°, Y axis angle > 68.9°, FHI > 0.68.

Normal: 27.3° ≤ SN-MP ≤ 37.7°, 25.5° ≤ FH-MP36.7°, 62.7° ≤ Y axis angle ≤ 68.9°, 0.62 ≤ FHI ≤ 0.68.

High: SN-MP > 37.7°, FH-MP > 36.7°, Y axis angle < 62.7°, FHI < 0.62.

### 2.2. Imaging Acquisition and Evaluation

The subjects all held the natural head position. All scans were performed by the same technician and were set at the following scanning parameters: FSV, 110 kV; 3.12 mA; voxel size, 0.3 mm; exposure time, 3.6 s; field of view, 18 × 16 cm. CBCT scans were exported in the digital imaging and communications in medicine (DICOM) format. The same device (Orthoceph OP300, Instrumentarium, Tuusula, Finland) was used to determine the vertical growth patterns of subjects performed on LCRs.

### 2.3. Landmarks and Definition of Sella Turcica and STB

Reference planes and the definitions of linear dimensions were identified to perform the analysis (Table 1).

We reoriented the three-dimensional images using reference planes and carried out reconstruction in Mimics (version 21.0, Materialise, Leuven, Belgium). According to methods proposed by Ortiz [18] and Ugurlu [24], the sella turcica was measured, as shown in Figure 1, Figure 2 and Figure 3.

Referring to the criteria given by Ortiz [18], in this study, STB was classified into two groups according to the ratio of interclinoid distance (ACP-PCP) to length (TS-DS), i.e., no bridging (ratio ≥ 60%) and bridging (ratio < 60%).

### 2.4. Examiner Reliability

To minimize any method errors that may occur in the study, 60 samples were randomly re-evaluated by the same researcher two weeks apart. Intraclass correlation coefficients (ICCs) were calculated, and the method errors were evaluated by Dahlberg’s formula: $\sqrt{d^{2}/2n}$ (d represents the deviations between the two measurements; n represents the number of paired objects).

### 2.5. Statistical Analysis

All the variables are described by the mean, standard deviation, and percentages. The Kolmogorov–Smirnov test and Levene’s test were performed to determine the distribution of normality and homogeneity of variances, respectively. In three groups, any difference in the ANB, SN-MP, FH-MP, Y axis angle, FHI, and ages of subjects was obtained using one-way analysis of variance. Student’s *t* test and the Mann–Whitney U test were used to assess the possible diversity in genders. To investigate the relationship between the sella turcica and different vertical growth patterns, one-way analysis of variance and the least significant difference test were applied. The prevalence of STB among different vertical growth patterns was compared using the chi-square test. Moreover, Pearson and Spearman correlations were used to evaluate the correlativity between the sella turcica and the cephalometric measurements that reflect vertical growth patterns for all subjects. The confidence interval was set as 95% and statistical significance was accepted at *p* < 0.05. Statistical analyses were conducted using SPSS (version 26.0, mac OS; IBM, Armonk, NY, USA).

## 3. Results

According to ICC (0.942–0.998), intra-observer reliability was excellent, and Dahlberg’s formula showed that the method errors ranged from 0.00 to 0.13 mm. The mean age, cephalometric measurements, and sex component ratio are shown in Table 2. The sex constituent ratio was consistent. The ANB angle showed that all subjects were skeletal Class I, and a strong agreement was found in the groups. Differences in the SN-MP, FH-MP, Y axis angle, and FHI were highly significant, confirming that they belong to different vertical growth patterns.

### 3.1. Different Vertical Growth Patterns with Different Sella Turcica Shapes

One hundred and twenty subjects were evaluated in this study. Table 3 shows the mean and standard deviation of sella turcica dimensions.

In terms of the dimensions of the sella turcica, among different vertical growth patterns (Table 4), statistical differences were recorded (*p* < 0.05). In the low-angle group, the posterior clinoid distance was larger, and the posterior left clinoid height, posterior right clinoid height, tuberculum sellae height, and dorsum sellae height were significantly smaller in the high-angle group, but we could not detect any difference between the normal-angle group and the high-angle group. Further correlation analysis shows that the posterior clinoid distance and the left and right anterior clinoid height were significantly correlated with all four vertical indicators (Table 5).

### 3.2. There Is Nearly No Difference in Sella Turcica Dimensions between Genders

Table 6 shows the differences in dimensions between genders and shows that the mean values were similar. Only slight increases in the anterior and posterior clinoid distance were found in males (*p* < 0.05). The mean differences and significance levels between the genders within three groups (Table 7 and Table 8) reported the same results.

### 3.3. Different Vertical Growth Patterns with Different Incidence of Sella Turcica Bridging

Table 9 shows that significant differences (χ^2^ = 10.00, *p* < 0.05) in the percentage of STB among different vertical growth patterns were found by the chi-square test, but no correlation existed in genders (χ^2^ = 0.556, *p* > 0.05). Compared with normal-angle (30%) and high-angle subjects (30%), STB frequency was found to be significantly higher in low-angle subjects (60%). However, no statistical differences were observed between normal-angle and high-angle subjects.

## 4. Discussion

The sella turcica was derived from the neural crest cells and the notochord mesodermal cells. The variation in shapes is affected by the development of the pituitary gland, which has an impact on the growth and development of individuals [5].

Many scholars have studied lengths, depths, and diameters as important indexes of sella turcica size. In our study, the mean length, depth, and diameter of the sella turcica were 10.01 ± 1.47 mm, 7.71 ± 1.30 mm, and 11.36 ± 1.43 mm, respectively. These results are consistent with studies by Sathyanarayana [25], who studied the mean length, depth, and diameter of the sella turcica in the South Indian race (9.6 ± 1.57 mm, 7.5 ± 1.36 mm, and 11.3 ± 1.25 mm, respectively). Compared with our study, the mean Saudi size was larger in the studies conducted by Alkofide EA (11.3 ± 2.58 mm, 9.3 ± 1.31 mm, and 14.5 ± 2.01 mm) [2], and the mean sella turcica size of Nepali citizens was smaller in the studies carried out by Shrestha (8.13 ± 2.03 mm, 9.60 ± 1.43 mm, and 6.40 ± 1.21 mm) [13]. This may be attributed to the difference in ethnicities, genes, and environmental factors.

When sella turcica shapes were compared in genders, Muhammed [26] and Sathyanarayana et al. [25] found that a significant difference was recorded in length using the lateral cephalogram. Contrary to these results, our study found a major agreement between genders, and the same results were also found by Islam [27] and Hasan [28], whose studies were based on CBCT. These differences can be attributed to the samples’ age, different imaging uses, or measurement errors, indicating that these influences cannot be ignored.

Axelsson et al. pointed out that no obvious change occurred in the dimensions of the sella turcica after the pubertal period [8,19]. Compared to Muhammed (8–28 years) [26] and Sathyanarayana (6–17 years) [25], our study enrolled subjects older than 16 years whose development had nearly stopped and whose sella turcica dimensions were stable. Hence, the confounding factor of age was avoided. Previous studies found that the overall prevalence of STB in the population ranged from 1% to 81%. In addition, a recent systematic review revealed that the large differences in prevalence might be due to measurement errors [16], leading to publication bias. The effect of the superimposition of anatomical structures and images was also confirmed, which made it difficult to discriminate between true bridging and pseudo bridging on lateral cephalograms [20,21]. Three-dimensional imaging, similar to CBCT, can accurately assess sella turcica shapes and is largely free of false-positive results [18]. The method judgment of STB also resulted in various differences. Becktor et al. [29] divided STB into two types: obvious band fusion and front–back or thin middle fusion. Leonardi [30] defined the radiographically visible diaphragm sella as STB, according to the calcification degree of the intercostal ligament. To overcome subjectivity in evaluating STB-related calcification, an objective quantitative method was performed in our study. We quantified the calcification degree present in the right and left clinoid processes and used the ratio [18] of interclinoid distance (ACP-PCP) to length (TS-DS) to differentiate whether there was STB or not. Moreover, more than ten CBCT measurement indexes, such as the anterior clinoid height, posterior clinoid height, tuberculum sellae height, dorsum sellae height, and so on, were added to measure the sella turcica with greater accuracy.

In our study, only the anterior clinoid distance and posterior clinoid distance showed slight gender differences, and no gender difference was found in the other indexes. Moreover, it is the first study that focuses on evaluating sella turcica shapes in different vertical growth patterns and genders.

Measuring the sella point as the marker could reflect the development and deformity of the mandibular [2]. In the early stage of orthodontic development, Bjork and Jarabak introduced the saddle angle (N-S-Ar) and articular angle (S-Ar-Go’) concepts to describe the position between the cranial base and facial bones in sagittal and vertical directions [31]. Recent studies correlated the sella turcica with different sagittal growth patterns using the lateral cephalogram to assess whether there is a relationship [12,32]. Investigations between the sella size and both Class II and Class III malocclusions were performed by Sathyanarayana [25] and Shrestha [13], and they found that Class II subjects had smaller sella turcica sizes than Class III subjects. In terms of diameter, Alkofide EA [2] noted that there were more Class III subjects than Class II subjects. Furthermore, in the study by Abdel Kaber [20], patients treated with combined orthodontics and orthognathic treatment had a 10.7% incidence of STB, compared to 7.1% for patients treated with orthodontics alone. Hence, he thought that STB could reflect the degree of jaw deformity. Dasgupta [33] showed that the incidence of STB was significantly higher in Class II patients. Some studies have also pointed out a higher incidence of STB in skeletal Class III subjects [7,34], and the study by Alkofide [2] revealed a strong link between STB and sagittal malocclusion. Based on these findings, scholars believe that sella turcica shapes can be used to assess different sagittal growth patterns [33]. The sagittal growth pattern was reflected in the developmental differences in the jaw.

Several studies have pointed out that the anterior wall of the sella turcica and the frontonasal region are developed from the neural crest cells in embryology, while the posterior wall appears to be associated with the posterior cranial base, which is developed from the para-axial mesoderm [5,12]. The anterior and posterior cranial bases were attached to the maxilla and mandible through growth sutures and temporomandibular joints, respectively. Therefore, any change in the cranial base can have an impact on the development of (and changes in) facial bones [35], not only including sagittal changes but also vertical growth changes. Driven by the vertical growth of the cranial base, vertical growth patterns will change. Consequently, further research on different vertical growth patterns will help to inform better orthodontic treatment decisions. The length, diameter, depth, and cross-sectional area of the sella turcica in different vertical growth patterns in Caucasian subjects have been evaluated via lateral cephalograms [15], but the link between them and STB has not been clarified. With all of this in mind, the relationship between the sella turcica and different vertical growth patterns was analyzed in this study. Statistical differences were observed in some indexes. A higher posterior clinoid distance and smaller posterior clinoid height were determined in the low-angle group, and the occurrence of STB was correlated with vertical growth patterns. A significantly higher incidence of STB was observed in the low-angle subjects.

These results indicate that differences in the sella turcica influenced vertical growth patterns, mostly concerning the posterior clinoid process and the tuberculum sellae. This could be explained from two aspects, as studies on genes have demonstrated that the anterior wall, the low wall, and the posterior wall are located in different maxillofacial developmental fields, respectively, such as the frontonasal boundary area, the palatal boundary area, and the mandibular boundary area. They migrated from diverse sources of cells and had different genetic backgrounds. The posterior wall shared a significantly similar origin with the posterior cranial base from the para-axial mesoderm. Some scholars, especially Kjaer, hold the view that changes in the posterior cranial base could affect the mandible [36] and the vertical growth. On the other hand, sella turcica shapes are closely related to the pituitary gland, which is located at the center of the sella turcica and secretes the important growth hormone. Some scholars have given growth hormone therapy to patients with growth retardation and found that growth hormone can significantly stimulate the growth of the mandible [37], which may be expressed as different vertical growth patterns in the craniofacial region. As a result, it is reasonable to assume that it might be possible to assess vertical trends by measuring the sella turcica.

It is important to understand sella turcica shapes. The sella point, as a cephalometric marker, can reflect the relative position of the maxilla and the mandible [2,38]. However, the role of the sella turcica, an indispensable structure in orthodontic diagnosis, is often ignored. Our study indicates that, in terms of the sella turcica, there were differences in vertical growth patterns, involving a smaller posterior clinoid height, a larger posterior clinoid distance, and a higher incidence of STB in low-angle subjects, which can be used as prompt information to assess vertical growth trends.

The equal sex distribution was considered in this study, but only the class I skeletal pattern was investigated. This limitation could not be ignored. As the research expands, these findings could be confirmed by establishing different vertical groups in Class II or Class III sagittal malocclusions.

Since the subjects were from our hospital rather than the general population, some inherent biases may exist. Although the present study avoids the limitations of 2d lateral cephalometric radiographs, as reported in the previous literature, a comparison of 2d and 3d analysis was not performed. Moreover, the small sample size and anthropological variations cannot be ignored. As the research expands, these findings could be confirmed by establishing a detailed 3d investigation using a larger sample size and a more comprehensive assessment of the sella turcica.

## 5. Conclusions

Genders were not linked to the sella turcica.Sella turcica shapes were associated with different vertical growth patterns, and low-angle subjects have a larger posterior clinoid distance, a smaller posterior clinoid height, and a higher incidence of sella turcica bridging.The posterior clinoid process and sella turcica bridging could be used as prompt information on vertical growth trends.

## Figures and Tables

**Figure 1 jcm-12-01890-f001:**
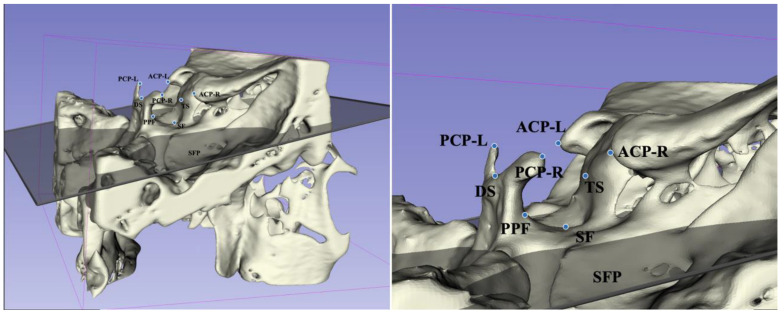
Sella turcica landmarks based on CBCT. ACP-R: the apex of the anterior clinoid process on the right side; ACP-L: the apex of the anterior clinoid process on the left side; PCP-R: the apex of the posterior clinoid process on the right side; PCP-L: the apex of the posterior clinoid process on the left side; DS: the dorsum sellae point; TS: tuberculum sellae; PPF: the most posterior point of the posterior wall of the sella turcica; SF: the deepest point of the sella turcica; SFP: sella floor plane.

**Figure 2 jcm-12-01890-f002:**
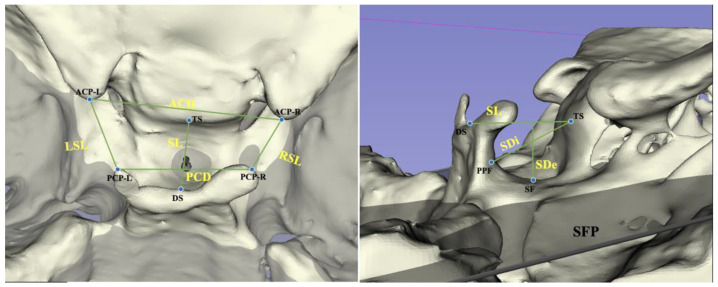
Sella turcica linear dimensions based on CBCT. ACD: anterior clinoid distance; PCD: posterior clinoid distance; LSL: left sella length; RSL: right sella length; SL: sella length; SDe: sella depth; SDi: sella diameter; ACP-R: the apex of the anterior clinoid process on the right side; ACP-L: the apex of the anterior clinoid process on the left side; PCP-R: the apex of the posterior clinoid process on the right side; PCP-L: the apex of the posterior clinoid process on the left side; DS: the dorsum sellae point; TS: tuberculum sellae; PPF: the most posterior point of the posterior wall of the sella turcica; SF: the deepest point of the sella turcica; SFP: sella floor plane.

**Figure 3 jcm-12-01890-f003:**
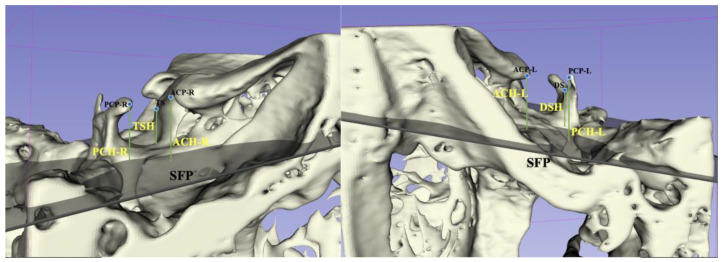
Sella turcica linear dimensions based on CBCT. TSH: tuberculum sellae height; PCH-R: posterior right clinoid height; ACH-R: anterior right clinoid height; PCH-L: posterior left clinoid height; ACH-L: anterior left clinoid height; DSH: dorsum sellae height; ACP-R: the apex of the anterior clinoid process on the right side; ACP-L: the apex of the anterior clinoid process on the left side; PCP-R: the apex of the posterior clinoid process on the right side; PCP-L: the apex of the posterior clinoid process on the left side; DS: the dorsum sellae point; TS: tuberculum sellae; PPF: the most posterior point of the posterior wall of the sella turcica; SFP: sella floor plane.

**Table 1 jcm-12-01890-t001:** Reference planes and the definitions of linear dimensions.

Item	Definition
Frankfort horizontal plane	The highest point on the upper margin of the opening of each external auditory canal and the low point on the lower margin of the left orbit
Midsagittal plane	A vertical longitudinal plane which is perpendicular to the Frankfort horizontal plane and passes through the N point and the Ba point
Sella floor plane	A plane that passes through the SF point and is parallel to the Frankfort horizontal plane
Anterior clinoid distance	The distance between ACP-R and ACP-L
Posterior clinoid distance	The distance between PCP-R and PCP-L
Left sella length	The distance between ACP-L and PCP-L
Right sella length	The distance between ACP-R and PCP-R
Sella length	The distance between TS and DS
Anterior left clinoid height	The vertical distance between ACP-L and the Frankfort horizontal plane moved to the sella floor plane
Anterior right clinoid height	The vertical distance between ACP-R and the Frankfort horizontal plane moved to the sella floor plane
Posterior left clinoid height	The vertical distance between PCP-L and the Frankfort horizontal plane moved to the sella floor plane
Posterior right clinoid height	The vertical distance between PCP-R and the Frankfort horizontal plane moved to the sella floor plane
Tuberculum sellae height	The vertical distance from the tuberculum sellae to the Frankfort horizontal plane moved to the sella floor plane
Dorsum sellae height	The vertical distance from the dorsum sellae to the Frankfort horizontal plane moved to the sella floor plane
Sella diameter	The distance between TS and PPF at the midsagittal plane
Sella depth	The vertical distance from SF to the connection between TS and DS at the midsagittal plane

**Table 2 jcm-12-01890-t002:** The mean age, cephalometric measurements, and sex distribution of each group.

	Age	ANB	SN-MP	FH-MP	*y* Axis Angle	FHI	Female		Male	
							N	%	N	%
Low angle	21.75	2.61	22.24	18.20	75.40	0.62	20	50	20	50
Norm angle	22.98	2.64	32.42	28.73	67.70	0.67	20	50	20	50
High angle	22.00	2.82	42.71	41.39	61.45	0.73	20	50	20	50
*p*	0.47	0.63	0.00 **	0.00 **	0.00 **	0.00 **	-	-	-	-

N, number of subjects; %, percentage; ** *p* <  0.01.

**Table 3 jcm-12-01890-t003:** The mean and standard deviation of sella turcica dimensions.

	Mean ± Standard Deviation	Range
Anterior clinoid distance	25.11 ± 2.46	19.30–34.98
Posterior clinoid distance	16.88 ± 2.59	10.95–24.36
Left sella length	7.00 ± 2.03	0–13.54
Right sella length	6.59 ± 2.02	0–10.92
Sella length	10.01 ± 1.47	5.51–13.80
Anterior left clinoid height	7.12 ± 1.62	3.38–12.11
Anterior right clinoid height	7.38 ± 1.72	1.79–12.04
Posterior left clinoid height	7.33 ± 2.05	2.67–14.54
Posterior right clinoid height	7.62 ± 1.88	3.06–14.74
Tuberculum sellae height	8.36 ± 1.40	4.87–13.58
Dorsum sellae height	7.09 ± 1.53	3.77–11.33
Sella diameter	11.36 ± 1.43	7.86–18.14
Sella depth	7.71 ± 1.30	4.90–11.40

**Table 4 jcm-12-01890-t004:** Differences in sella turcica dimensions among three groups.

	Value			*p*
	Low	Normal	High	
Anterior clinoid distance	25.31 ± 2.62	25.25 ± 2.53	24.79 ± 2.23	0.586
Posterior clinoid distance	17.78 ± 2.75 ^c^	16.67 ± 2.27	16.21 ± 2.53 ^a^	0.019 *
Left sella length	6.63 ± 1.87	7.09 ± 2.40	7.27 ± 1.76	0.346
Right sella length	6.10 ± 2.08	6.80 ± 1.96	6.86 ± 1.98	0.178
Sella length	10.12 ± 1.47	9.91 ± 1.54	10.01 ± 1.42	0.805
Anterior left clinoid height	6.96 ± 1.75	6.94 ± 1.32	7.47 ± 1.73	0.258
Anterior right clinoid height	7.39 ± 1.75	7.22 ± 1.48	7.53 ± 1.93	0.734
Posterior left clinoid height	6.44 ± 1.33 ^b,c^	7.64 ± 2.22 ^a^	7.91 ± 2.22 ^a^	0.003 **
Posterior right clinoid height	6.68 ± 1.18 ^b,c^	7.94 ± 1.83 ^a^	8.23 ± 2.15 ^a^	0.000 **
Tuberculum sellae height	8.15 ± 1.59 ^c^	8.08 ± 1.19 ^c^	8.86 ± 1.30 ^a,b^	0.021 *
Dorsum sellae height	6.53 ± 1.01 ^b,c^	7.49 ± 1.56 ^a^	7.23 ± 1.79 ^a^	0.015 *
Sella diameter	11.27 ± 1.78	11.34 ± 1.33	11.47 ± 1.11	0.807
Sella depth	7.83 ± 1.24	7.54 ± 1.21	7.77 ± 1.44	0.577

^a^ Difference with the low-angle group; ^b^ Difference with the normal-angle group; ^c^ Difference with the high-angle group; * *p* < 0.05, ** *p* < 0.01.

**Table 5 jcm-12-01890-t005:** Correlation between sella turcica dimensions and cephalometric measurements.

	SN-MP		FH-MP		Y Axis Angle		FHI	
	*r*	*p*	*r*	*p*	*r*	*p*	*r*	*p*
Posterior clinoid distance	−0.241 **	0.008	−0.221 *	0.015	−0.219 *	0.016	0.272 **	0.003
Posterior left clinoid height	0.325 **	0.000	0.287 **	0.001	0.209 *	0.022	−0.287 **	0.002
Posterior right clinoid height	0.378 **	0.000	0.316 **	0.000	0.287 **	0.002	−0.362 **	0.000
Tuberculum sellae height	0.264 **	0.004	0.208 *	0.022	0.121	0.189	−0.304 **	0.001
Dorsum sellae height	0.198 *	0.030	0.194 *	0.033	0.151	0.101	−0.170	0.064

* *p* < 0.05, ** *p* < 0.01; *r*, the value for Pearson and Spearman correlation test.

**Table 6 jcm-12-01890-t006:** Differences in sella turcica dimensions between genders.

	Mean ± Standard Deviation		*p*
	Male	Female	
Anterior clinoid distance	26.10 ± 2.59	24.13 ± 1.87	0.000 **
Posterior clinoid distance	17.65 ± 2.79	16.13 ± 2.14	0.001 **
Left sella length	7.27 ± 2.22	6.72 ± 1.80	0.143
Right sella length	6.62 ± 2.01	6.55 ± 2.05	0.858
Sella length	10.12 ± 1.49	9.91 ± 1.45	0.428
Anterior left clinoid height	7.09 ± 1.79	7.14 ± 1.45	0.846
Anterior right clinoid height	7.23 ± 1.98	7.53 ± 1.41	0.348
Posterior left clinoid height	7.41 ± 2.14	7.25 ± 1.98	0.657
Posterior right clinoid height	7.63 ± 1.95	7.61 ± 1.82	0.976
Tuberculum sellae height	8.20 ± 1.56	8.52 ± 1.22	0.213
Dorsum sellae height	7.07 ± 1.35	7.10 ± 1.71	0.922
Sella diameter	11.28 ± 1.64	11.44 ± 1.18	0.559
Sella depth	7.62 ± 1.26	7.81 ± 1.33	0.417

** *p* < 0.01.

**Table 7 jcm-12-01890-t007:** Mean and standard deviation of sella turcica dimensions in genders within three groups.

	Low Angle		Normal Angle		High Angle	
	Male	Female	Male	Female	Male	Female
Anterior clinoid distance	26.26 ± 2.80	24.36 ± 2.08	26.29 ± 2.85	24.20 ± 1.65	25.75 ± 2.15	23.83 ± 1.91
Posterior clinoid distance	18.80 ± 2.78	16.77 ± 2.38	17.09 ± 2.76	16.25 ± 1.60	17.05 ± 2.58	15.37 ± 2.22
Left sella length	6.55 ± 2.03	6.71 ± 1.73	7.60 ± 2.73	6.42 ± 1.84	7.66 ± 1.71	6.88 ± 1.76
Right sella length	6.03 ± 2.23	6.17 ± 1.98	7.17 ± 2.06	6.42 ± 1.84	6.66 ± 1.65	7.06 ± 2.29
Sella length	10.33 ± 1.64	9.91 ± 1.30	10.14 ± 1.35	9.68 ± 1.71	9.90 ± 1.51	10.13 ± 1.34
Anterior left clinoid height	6.58 ± 1.81	7.33 ± 1.64	7.20 ± 1.32	6.68 ± 1.31	7.50 ± 2.10	7.43 ± 1.32
Anterior right clinoid height	7.07 ± 1.78	7.71 ± 1.71	7.35 ± 1.78	7.10 ± 1.14	7.28 ± 2.41	7.78 ± 1.30
Posterior left clinoid height	6.47 ± 1.35	6.41 ± 1.34	7.88 ± 2.29	7.40 ± 2.18	7.89 ± 2.40	7.92 ± 2.10
Posterior right clinoid height	6.65 ± 1.41	6.71 ± 0.94	8.25 ± 1.59	7.63 ± 2.03	7.97 ± 2.40	8.50 ± 1.89
Tuberculum sellae height	7.73 ± 1.84	8.57 ± 1.19	8.16 ± 1.23	7.99 ± 1.17	8.71 ± 1.47	9.01 ± 1.13
Dorsum sellae height	6.27 ± 1.04	6.81 ± 0.93	7.82 ± 0.99	7.17 ± 1.95	7.14 ± 1.51	7.32 ± 2.06
Sella diameter	11.10 ± 2.25	11.43 ± 1.18	11.50 ± 1.28	11.17 ± 1.38	11.24 ± 1.25	11.79 ± 0.94
Sella depth	7.56 ± 1.42	8.11 ± 0.98	7.73 ± 0.88	7.36 ± 1.47	7.57 ± 1.46	7.96 ± 1.43

**Table 8 jcm-12-01890-t008:** Mean differences and significant levels between genders within three groups.

	Value			*p*		
	Low	Normal	High	Low	Normal	High
Anterior clinoid distance	1.90 ± 0.78	2.09 ± 0.74	1.92 ± 0.64	0.020 *	0.007 **	0.005 **
Posterior clinoid distance	2.03 ± 0.82	0.84 ± 0.71	1.68 ± 0.76	0.018 *	0.247	0.034 *
Left sella length	−0.16 ± 0.60	1.02 ± 0.75	0.78 ± 0.55	0.792	0.185	0.164
Right sella length	−0.15 ± 0.67	0.74 ± 0.62	−0.40 ± 0.63	0.829	0.236	0.530
Sella length	0.41 ± 0.47	0.46 ± 0.49	−0.24 ± 0.45	0.380	0.351	0.605
Anterior left clinoid height	−0.75 ± 0.55	0.52 ± 0.42	0.07 ± 0.56	0.175	0.222	0.906
Anterior right clinoid height	−0.64 ± 0.55	0.25 ± 0.47	−0.50 ± 0.61	0.254	0.596	0.417
Posterior left clinoid height	0.06 ± 0.42	0.48 ± 0.71	−0.04 ± 0.71	0.895	0.500	0.960
Posterior right clinoid height	−0.06 ± 0.38	0.29 ± 0.62	−0.53 ± 0.68	0.875	0.290	0.444
Tuberculum sellae height	−0.84 ± 0.49	0.65 ± 0.17	−0.30 ± 0.41	0.096	0.654	0.480
Dorsum sellae height	−0.54 ± 0.31	0.20 ± 0.65	−0.19 ± 0.57	0.091	0.196	0.748
Sella diameter	−0.33 ± 0.57	0.43 ± 0.33	−0.46 ± 0.35	0.561	0.433	0.196
Sella depth	−0.55 ± 0.39	0.34 ± 0.37	−0.39 ± 0.46	0.159	0.343	0.396

* *p* < 0.05, ** *p* < 0.01

**Table 9 jcm-12-01890-t009:** Difference in STB in genders and three groups.

	None STB		STB		Statistics ^a^	
	N	%	N	%	Value	*p*
Female	38	63.3	22	36.7	0.556	0.456
Male	34	56.7	26	43.3		
Low angle	16 ^c,d^	40	24 ^c,d^	60	10.00	0.007 **
Normal angle	28 ^b^	70	12 ^b^	30		
High angle	28 ^b^	70	12 ^b^	30		

^a^ Chi-square test; ^b^ Difference with the low-angle group; ^c^ Difference with the normal-angle group; ^d^ Difference with the high-angle group; ** *p* < 0.01.

## Data Availability

All data included in the results can be obtained by contacting the corresponding author.

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
