# Peer review of "A CBCT Investigation of the Sella Turcica Dimension and Sella Turcica Bridging in Different Vertical Growth Patterns"

_jcm, 2023, doi:10.3390/jcm12051890_

Round 1
Reviewer 1 Report
Presented study is an example of well prepared and properly conducted scientific resarch.
As a reader I had a problem in Introduction with a sentences:
Moreover, the mutations of the homeobox genes in cranial neural crest cells may have an impact on the development of the dentition and midface through signaling conduction[6]. This suggested that the sella turcica is closely related to the growth and development of craniofacial region.
I prefer to read:
This could suggested that the sella turcica morphology is closely related to the growth and development of craniofacial region.
I think that it is only language correction.
Also in conclusions lack of abbreviations would be prefarable.
Reviewer 2 Report
Please explain the reasoning ( justification) for lateral cephalograms. This is very important based on the ALARA rule, it is not clear why additional X-rays were performed.
There are no clinical or other benefits of the article reported eg. The mean sella turcica dimensions may be used as reference standards in future 308 for Chinese subjects when studying sella turcica morphology - which as you have a different skeletal growth pattern? the sample size is small to have this conclusion.
No limitation of the study is included in the article such as a comparison of 2d and 3d analysis (LCG and CBCT), sample size, left-handers /right-handers, anthropological variations of sella etc.
Author Response
Thank you so much for your great efforts and approval.
Please see the attachment.

Round 2
Reviewer 2 Report
Thank you, for clarifying all raised issues. Have many citations!